# Invariant and smooth limit of discrete geometry folded from bistable origami leading to multistable metasurfaces

Ke Liu [1], Tomohiro Tachi[2] & Glaucio H. Paulino [3]

Origami offers an avenue to program three-dimensional shapes via scale-independent and non-destructive fabrication. While such programming has focused on the geometry of a tessellation in a single transient state, here we provide a complete description of folding smooth saddle shapes from concentrically pleated squares. When the offset between square creases of the pattern is uniform, it is known as the pleated hyperbolic paraboloid (hypar) origami. Despite its popularity, much remains unknown about the mechanism that produces such aesthetic shapes. We show that the mathematical limit of the elegant shape folded from concentrically pleated squares, with either uniform or non-uniform (e.g. functionally graded, random) offsets, is invariantly a hyperbolic paraboloid. Using our theoretical model, which connects geometry to mechanics, we prove that a folded hypar origami exhibits bistability between two symmetric configurations. Further, we tessellate the hypar origami and harness its bistability to encode multi-stable metasurfaces with programmable non-Euclidean geometries.

[1] Department of Mechanical and Civil Engineering, California Institute of Technology, Pasadena, CA 91125, USA. [2] Graduate School of Arts and Sciences, University of Tokyo, Tokyo, Japan. [3] School of Civil and Environmental Engineering, Georgia Institute of Technology, 30332 Atlanta, Georgia. Correspondence and requests for materials should be addressed to G.H.P. (email: paulino@gatech.edu)

The goal of programming non-Euclidean geometry from flat sheets is to control its shape, which usually involves a mix of art and technology. For instance, in thin shells, local geometric incompatibilities lead to out-of-plane buckling to create various three-dimensional shapes. Such phenomena abound in nature, e.g., wavy flowers and leaves[1,2], the growth pattern of guts[3], and the wrinkles on our brains[4]. Persistent efforts have been made to harness the buckling of thin shells into non-Euclidean shapes toward a range of applications[5–7]. Although the underlying mechanism is different, principles of origami (folding) can also be used to transform thin flat sheets into non-Euclidean shapes through purely isometric deformations at the local level (i.e., no stretching or cutting)[8–13]. For example, origami has been used to achieve structures with various interesting mechanical properties, such as tunable Poisson's ratio[14–16], programmable stiffness[17–20], and multi-stability[21–25]. As a result, origami has led to major advances in multi-functional structures and programmable metamaterials[26–28].

While it is natural to imagine smooth surfaces arising from curved folds[11,13,29]; discrete folds can also converge to smooth surfaces, which has been demonstrated using the Miura tessellation[8] and the pleated hyperbolic paraboloid (hypar) origami[30–32]. Here we focus on the latter, which is obtained by folding a piece of paper along concentric squares and their diagonals to arrive at a seemingly smooth saddle shape. Unlike conventional origami patterns used to generate non-Euclidean shapes[8–10], the hypar origami does not arise from a tessellated pattern, and thus has no periodic unit cells. The hypar origami is popular in decorative arts, owing to its aesthetic shape, simple pattern, and its tolerance to geometric variations, as demonstrated in Fig. 1. The hypar pattern possesses attractive mechanical properties, such as structural bistability[33,34]. Interestingly, the two stable configurations of a hypar origami are symmetric to each other, which makes it a promising platform for multi-functional devices or metasurfaces.

Despite its aesthetic appeal and potential for technological applications, a comprehensive understanding of the hypar origami remains elusive because of the challenges associated with it being a non-rigid origami with non-periodic pattern. In an attempt to gain a deeper theoretical understanding of the hypar origami and use it for engineering applications, we aim to answer the following questions: can we prove, theoretically, that the actual folded shape of the hypar origami is a hyperbolic paraboloid? How does the local deformation of each panel relates to the global shape of the pattern? What are the conditions for the

bistability to exist? How can we use the hypar pattern to create metasurfaces?

In what follows, we develop a theoretical model of the geometry and mechanics of the pleated hyperbolic paraboloid (hypar) origami, and compare the results with numerical simulations and experiments. Instead of looking at certain transient states, we construct a complete analytical description of folding concentric squares by homogenizing local folds to establish a differential map of the global geometry. By solving the obtained differential equation, we show that the analytical limit of the folded shape of the hypar origami, at any stage of its folding process, is indeed a hyperbolic paraboloid, as its name suggests. We also show that the same geometric limit holds true for general pleated concentric squares with or without uniform offsets, such as the ones shown in Fig. 1. Our analytical description explicitly connects the global curvature and local folds, and helps us to connect the mechanics of the hypar pattern with its geometry. It is followed by a proof that the bistability of the hypar pattern always exist when a few basic assumptions are satisfied. We build both physical models and numerical models to verify the analytical model. Although established based on simplifications, the analytical model is able to accurately predict the actual geometry and mechanical behavior of the hypar pattern. For the numerical model, a bar-and-hinge reduced order model is used to conduct nonlinear simulations, capturing the bistable snapping between two symmetric stable states of folded hypar structures. Finally, using this knowledge, we create a mechanical metasurface by tessellating the hypar pattern to achieve programmable non-Euclidean geometries.

## Results
**Analytical limit of the folded geometry.** Once folded, the hypar pattern buckles out of the plane into a saddle-shaped shell to resolve the fundamental incompatibility between the in-plane strain induced by folding and its three-dimensional embedding[32]. Each panel in the pattern is subjected to twisting along its longitudinal direction[32]. Performing a simple surgery of cutting one corrugation out, we observe an immediate release of the twist, causing the corrugation to lose compatibility as we try to fit it back onto the folded shell, as demonstrated by the insets in Fig. 2a, b. The folded hypar sheet has two orthogonal symmetry planes spanned by the diagonal creases, which divide the shell into four symmetric pieces, each within a quadrant of the *x, y*-plane. From a homogenized view of the global deformation, we

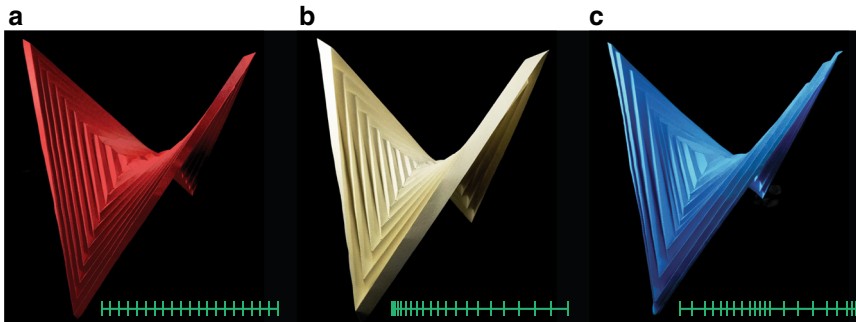

**Fig. 1** The folded shapes of concentrically pleated square (and diagonals) models made by paper. From a far distance, the three structures look similar; however, as we zoom in, we notice differences in their local patterns. **a** The saddle shape folded from the standard hypar pattern with uniform offsets between squares, as indicated by the green marks. **b** The similar saddle shape folded from a functionally graded hypar pattern with increasing offsets from the center to the outside. **c** The similar saddle shape folded from a random hypar pattern with random offsets between square creases. Perhaps this explains the popularity of the hypar origami: it is beautiful, simple, and tolerates local geometrical variations (e.g., uniform, graded, and random offsets); however, the global shape displays geometric invariance

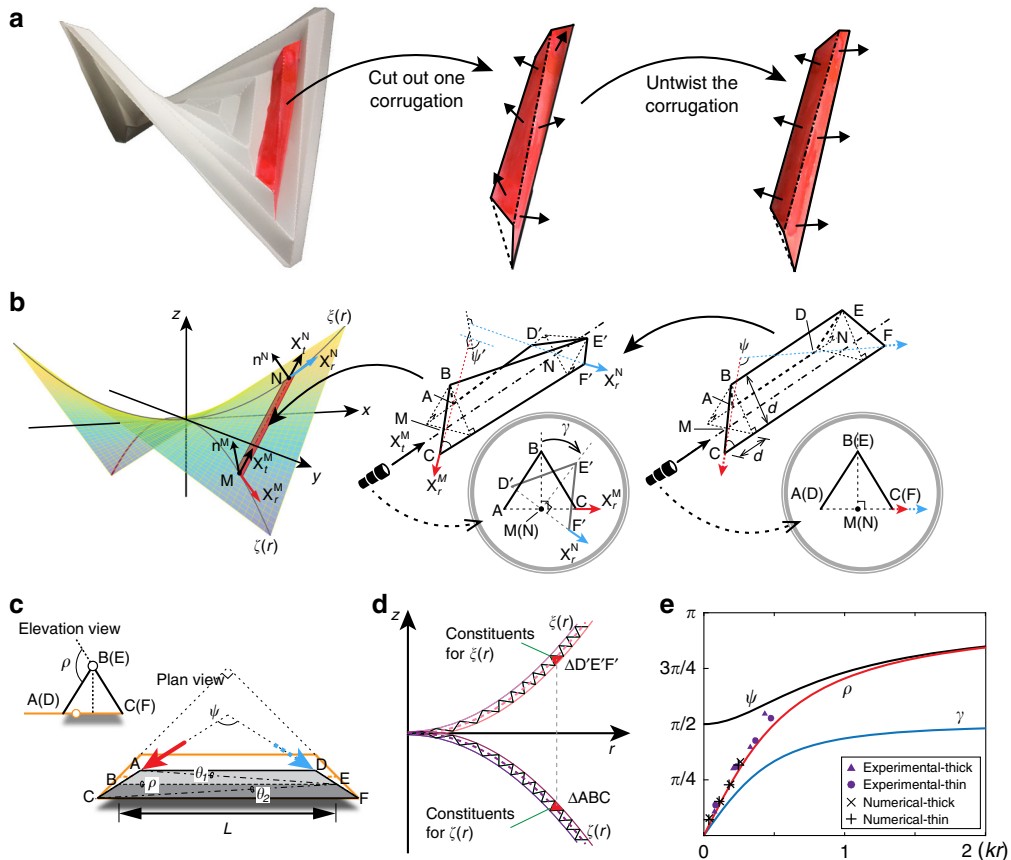

**Fig. 2** Schematic of the global and local configurations of a hypar origami. **a** A surgery on the hypar origami takes out a twisted corrugation, which untwists into a simple straight fold. The black arrows indicate surface normals. **b** We describe the global saddle shape of a hypar folded shell by the union of four pieces of ruled surface subject to reflection symmetry. Each corrugation resembles a ruling fiber. A folded corrugation must be twisted to satisfy the global compatibility constraint. The circular insets show a projection view looking through the longitudinal axis of a corrugation. **c** Plan and elevation views of a folded corrugation before twisting. The folding angle $\rho$, and two bending angles $\theta_1, \theta_2$ are labeled. **d** The construction of curves $\xi$ and $\zeta$. The black lines show the folded diagonal creases of the hypar origami. **e** The analytical curves that relate the global geometry of a hypar origami measured by $kr$ with the local geometry of a corrugation measured by the folding angle $\rho$, opening angle $\psi$, and twisting angle $\gamma$. Experimental and numerical data are sampled from the scanned and simulated models, respectively

define a surface that describes the shape of the corrugated shell. We cannot assume smoothness at the joint between any two pieces of surface from adjacent quadrants due to the inherent discrete nature of the square hypar. Nevertheless, within each quadrant, the piece of surface is supposed to be a smooth ruled surface bounded by two curves, which can be parametrized as:

$$\mathbf{X}(r,t) = (1-t)\zeta(r) + t\xi(r), \quad 0 \le t \le 1, \tag{1}$$

where $\zeta(r)$ and $\xi(r)$ are curves lying on the symmetry planes constructed approximately by the folded diagonal creases. The straight corrugations resemble rulings.

The hypar origami is not rigid foldable based on its original pattern[31]. To analytically describe the geometry of each deformed corrugation, we assume that the deformations are isometric, and straight creases remain straight, which requires the introduction of at least one additional diagonal pleat in each trapezoidal panel to triangulate the pattern[35,36]. We adopt the alternating asymmetric triangulation of Demaine et al.[31] as it satisfies the reflection symmetry required by our present analytical model (see Supplementary Fig. 1). We parametrize a corrugation by one folding angle $\rho \in [0, \pi]$ and two bending angles $\theta_1, \theta_2 \in [0, \pi]$, considering panel width $d$ and folding ridge length $L$, as shown in Fig. 2c. The dimensionless width $w = (d/L)$ of a corrugation quickly vanishes away from the center. Denoting $n$ as the number of square creases counted from the center, for a standard hypar

pattern, we see that $(d/L) = 1/(2n)$, regardless of the actual dimension of the pattern. In the limit of $w \to 0$, the twisting angle $\gamma$ of a long corrugation becomes the average of $\theta_1$ and $\theta_2$. Due to orthogonality of the two symmetry planes, the virtual faces $\Delta ABC$ and $\Delta D'E'F'$ must be perpendicular to each other (see Fig. 2). Compatibility in radial directions further requires that $\theta_1 = \theta_2$. Detailed derivations are presented in Supplementary Note 1. Thus we have

$$\lim_{w \to 0} \gamma = \theta_1 = \theta_2, \text{ and } \lim_{w \to 0} \cos\gamma = \cos^2(\rho/2). \tag{2}$$

The fact that $\theta_1 = \theta_2$ leads to $\Delta ABC \cong \Delta D'E'F'$, which implies that $\zeta(r)$ and $\xi(r)$ have the same constituents at the outer rims, and thus we may assume that $\zeta(r)$ and $\xi(r)$ have the same shape, but opposite orientations (see Fig. 2d). Eventually, the twisted opening angle of a long corrugation ($\psi'$ in Fig. 2b) becomes

$$\lim_{w \to 0} \cos\psi' = \cos\gamma - 1. \tag{3}$$

We remark that the same identities resulting from an asymptotic analysis can be obtained using the other triangulation scheme, i.e. the asymmetric triangulation[31]. The same conclusion based on the two triangulation schemes reveals that there are fundamental constructions of the folded hypar pattern, which are independent of local deformations between creases (i.e., within

panels), as we take the limit to when each corrugation becomes a fiber on the homogenized surface.

Choosing a convenient coordinate system as shown in Fig. 2a, we simplify the parametrization of the surface in the first quadrant to:

$$\mathbf{X}(r, t) = [tr, (1 - t)r, (2t - 1)\xi(r)], \quad r \geq 0, \quad 0 \leq t \leq 1, \quad (4)$$

given $\boldsymbol{\xi}(r) = [r, 0, \xi(r)]$, where each entry denotes a Cartesian component of a vector in real space. When $w \to 0$, each corrugation becomes an infinitesimally thin ruling fiber connecting points M and N on the two bounding curves, respectively (see Fig. 2b). Thus we realize that

$$\frac{\mathbf{X}_r^M}{\|\mathbf{X}_r^M\|} \approx \frac{\mathbf{e}_{AC}}{\|\mathbf{e}_{AC}\|} \quad \text{and} \quad \frac{\mathbf{X}_r^N}{\|\mathbf{X}_r^N\|} \approx \frac{\mathbf{e}_{D'F'}}{\|\mathbf{e}_{D'F'}\|}, \quad (5)$$

where $\mathbf{e}_{IJ}$ denotes the vector from points I to J (I, J = A, C, D, F, D′, F′, see Supplementary Fig. 2). The twisting angle $\gamma$ of a corrugation is equal to the change of surface normal traveling along the corresponding ruling. Thus, we obtain two identities that relate the local geometry of corrugations with the global geometry of the folded shell:

$$\cos\gamma = \mathbf{n}^M \cdot \mathbf{n}^N, \quad \text{and} \quad \cos\psi' = \frac{\mathbf{X}_r^M \cdot \mathbf{X}_r^N}{\|\mathbf{X}_r^M\|\|\mathbf{X}_r^N\|}. \quad (6)$$

The surface tangents and normals can be derived by differentiating Eq. (4). Substituting Eq. (6) into Eq. (3), we reduce the problem of finding the shape of a surface to solving an ordinary differential equation (ODE) of the one dimensional function $\xi(r)$ (see Supplementary Note 2). The ODE reads

$$(2\xi(r) - r\xi'(r))(2r\xi'(r)^3 + 3r\xi'(r) - 2\xi(r)) = 0, \quad (7)$$

which is satisfied whenever either:

$$2\xi(r) - r\xi'(r) = 0, \quad (8)$$

or,

$$2r\xi'(r)^3 + 3r\xi'(r) - 2\xi(r) = 0. \quad (9)$$

If Eq. (8) is zero, we obtain the elegant solution:

$$\xi(r) = kr^2, \quad (10)$$

where $k$ is an arbitrary real constant. Thus we obtain the surface parametrization in the $x$, $y$, $z$ coordinates as:

$$\mathbf{X}(r, t) = [\pm tr, \pm(1 - t)r, (2t - 1)kr^2], \quad r \geq 0, \quad 0 \leq t \leq 1. \quad (11)$$

The signs of the $x$ and $y$ values have four different combinations, covering the four quadrants of the $x$, $y$-plane. Consequently, we can express the local folding angles as functions of $k$ and $r$, as shown in Fig. 2e.

**The unexpected solution**. A solution to Eq. (9) implies concavity for increasing $\xi$ and convexity for decreasing $\xi$, as shown in Fig. 3a, which does not agree with our observations of the standard hypar origami (that has a saddle shape). Interestingly, the unexpected shape is only achieved if we cut slits on the hypar pattern to make it a kirigami, as shown in Fig. 3b–d.

Differentiating Eq. (9) with respect to $r$, we obtain

$$\xi''(r) = -\frac{2\xi'(r)^3 + \xi'(r)}{3r(2\xi'(r)^2 + 1)}. \quad (12)$$

If $\xi'(r) \geq 0$, any real solution to Eq. (9) leads to $\xi'' \leq 0$, and thus we get a concave function when $\xi$ is increasing; on the other hand, $\xi'' \geq 0$ if $\xi'(r) \leq 0$, and thus the function must be convex when $\xi$ decreases. We can solve the ODE in Eq. (9) numerically. An example for $\xi(1) = 1$ is shown in Fig. 3a. Indeed, this solution

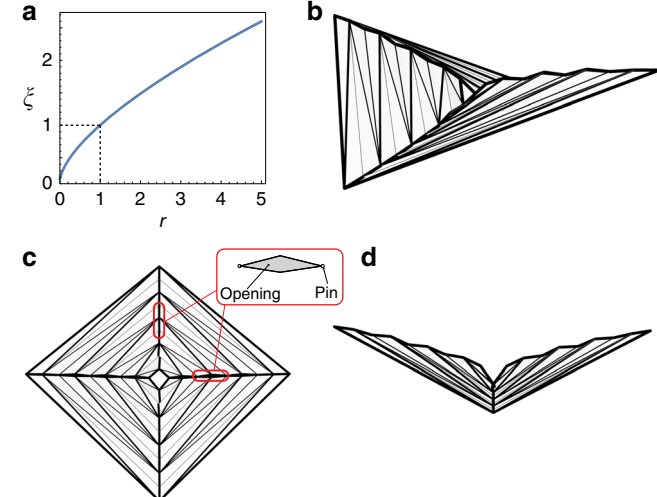

**Fig. 3** Solution to ODE Eq. (9), and its realization. **a** A solution to Eq. (9) solved numerically by assuming $\xi(1) = 1$. **b–d** By cutting slits on the hypar crease pattern to make it a kirigami, we can realize the solution to Eq. (9), which satisfies Eq. (3), but breaks the compatibility constraints. **c** Top view of the folded hypar kirigami. Slits are cut along the diagonals. Detailed view of a slit is shown in the inset. **d** Side view of the folded hypar kirigami. Diagonal creases approximate the solution of $\xi$ given by Eq. (9)

arises from the fact that Eq. (3) is only a necessary condition to the compatibility constraints. It is possible to satisfy Eq. (3), without complying with the compatibility constraints that we considered, by cutting slits on the crease pattern to make it a kirigami. The hypar-based kirigami gains extra degrees of freedom in folding and allows Eq. (3) to be satisfied, as demonstrated in Fig. 3b. Therefore, a solution to Eq. (9) does not describe the shape of a naturally folded hypar origami.

**Invariance of the analytical limit**. The configuration of a folded hypar origami is drawn from Eq. (11), which is the solution of the differential Eq. (8). We use symmetry to join the four pieces of surface together—a serendipitous finding is that the surface tangents and normals on the joint curves are consistent for any two adjacent pieces. There is no kink on the entire surface of the folded hypar shell, from a global homogenized viewpoint. However, Eq. (11) is not intuitive to interpret, as $k$ and $r$ change simultaneously when tracking a point on the curve $\xi$. Therefore, we rewrite $\xi$ as a function of $k$ and $p$, where $p$ is the distance of a point from the pattern centroid in the initial flat configuration, which remains unchanged for each point on $\xi$ (see Supplementary Fig. 4). The new function, denoted as $\tilde{\xi}$, is given by (see Supplementary Note 3):

$$\tilde{\xi}(k, p) = \frac{\sqrt{8p^2k^2 + 1} - 1}{4k}. \quad (13)$$

Accordingly, the hyperbolic paraboloid surface can be reparametrized as:

$$\tilde{\mathbf{X}}(p, t) = \begin{bmatrix} \pm \frac{t}{2}\sqrt{\frac{\sqrt{8k^2p^2+1}-1}{k^2}}, \\ \pm \frac{1-t}{2}\sqrt{\frac{\sqrt{8k^2p^2+1}-1}{k^2}}, \\ (2t - 1)\frac{\sqrt{8p^2k^2+1}-1}{4k} \end{bmatrix}, \quad p \geq 0, \quad 0 \leq t \leq 1, \quad (14)$$

which maps $p$, $t$ to the three-dimensional $x$, $y$, $z$-space, given coefficient $k$. We find that $z = k(x^2 - y^2)$, indicating that the hypar origami folds asymptotically to a smooth surface of hyperbolic paraboloid, which is maintained along the whole

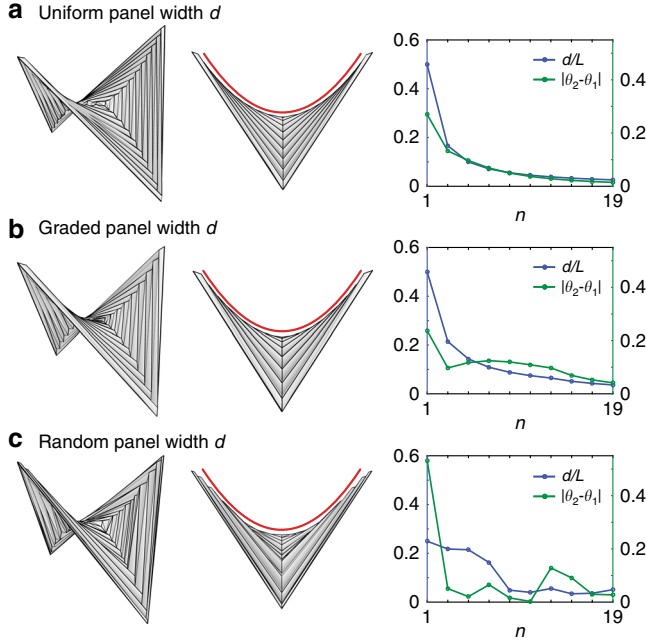

**Fig. 4** Folded configurations of three triangulated hypar origami with different panel widths display invariant hypar geometry (same patterns as in Fig. 1). The three origami patterns have the same number of square creases (counted from the center outwards), denoted by $n$. Every other square crease is a mountain, and thus there are 10 mountain creases in total. Two panels on the both sides of a mountain crease are assigned the same width. All three concentrically pleated origami can fold approximately into the same hypar shape, as indicated by the red quadratic curves. **a** Standard hypar pattern: panel width $d$ is a constant for all corrugations. **b** Functionally graded hypar pattern: panel width $d$ increases as $n$ increases. **c** Random hypar pattern: panel width $d$ is a random variable for each corrugation. Since the hypar patterns are triangulated, folding is achieved by rigid origami simulation[36]. The left images show the 3D views of the folded hypar origami. The middle images show the side views. The right images present quantitative measures. The blue dots (each dot corresponds to a mountain) in the charts show how $(d/L)$ vanishes as $n$ increases. The green dots show the residual (or error) of Eq. (2) when $n$ is finite

folding process. Intuitively, the more a hypar pattern is folded, the deeper the saddle, and the larger the $k$. This is quite a unique feature, as other approaches of folding a flat sheet into a hyperbolic paraboloid only guarantee to match the shape at a specific transient time during folding[8].

Moreover, the hyperbolic paraboloid shape is maintained as long as the dimensionless width $w$ of a corrugation vanishes away from the center, regardless of the actual value of $d$. We can assign non-uniform values to $d$, and still get a folded shape that approximates a hyperbolic paraboloid. In Fig. 4, we use three examples to illustrate this idea. In Fig. 4a, $d$ is uniformly assigned (standard hypar pattern); in Fig. 4b, $d$ is gradually increasing away from the center (functionally graded hypar pattern); in Fig. 4c, $d$ is randomly assigned by a uniform distribution between two bounding values, ensuring that $w_{max} = (d_{max}/L) \to 0$ (random hypar pattern).

## Connecting geometry with mechanics.

With an analytical description of the global homogenized geometry of the hypar pattern, we can associate the mechanical behavior of the pattern with its geometry by expressing the system energy in terms of the folding kinematics. First, let us consider how a hypar origami

folds into a stable saddle shape. We start from a flat hypar pattern, and forcefully fold up the pattern. After the origami is folded into shape, the folding creases undergo inelastic deformation, which shifts the neutral (i.e., stress free) angles of the folding hinges from zero to the angles at a folded state with shape coefficient $k = k_0$ (see Eq. 10). However, the panels undergo elastic deformation, as we observe that they return to the initial flat configuration once cut out from a folded hypar origami. Thus, we assume that the bending hinges still have their neutral angles equal to zero, which corresponds to the state at $k = 0$. A stable configuration is drawn by minimizing the system total energy ($E_T$).

Because we assume isometry for the analytical model, we have $E_T = E_F + E_B$, where $E_F$ denotes the folding energy and $E_B$ denotes the bending energy. To derive $E_F$ and $E_B$, we need to draw explicit maps between the global configuration and local angles. For instance, we obtain (see Supplementary Note 3):

$$\rho = \rho(k, p) = cos^{-1}\left(\frac{2}{\sqrt{8k^2p^2 + 1}} - 1\right), \quad (15)$$

$$\beta = \beta(k, p) = \pi - 2sin^{-1}\left(\frac{\sqrt{2}}{\sqrt{1 + \sqrt{8k^2p^2 + 1}}}\right), \quad (16)$$

$$\theta = \theta(k, p) = cos^{-1}\left(\frac{1}{\sqrt{8k^2p^2 + 1}}\right), \quad (17)$$

where the folding angles of the corrugation crease and the diagonal crease are denoted as $\rho$ and $\beta$, respectively (see Supplementary Fig. 3), and $\theta$ is the bending angle. Here we adopt the $k$, $p$-parametrization for ease of comparison between different folded states. We can then derive the folding and bending energy as:

$$E_F = 4\int_0^P \bar{\eta}\left[(\sqrt{2}p)\mathcal{H}_F^\rho + \mathcal{H}_F^\beta\right]dp, \quad (18)$$

$$E_B = 4\int_0^P \bar{\eta}(\sqrt{2}p)\mathcal{H}_B^\theta dp. \quad (19)$$

In Eq. (18), $P$ defines the size of the pattern, $\bar{\eta}$ is determined by the distribution of creases (see Supplementary Fig. 5), and $(\sqrt{2}p)$ refers to the length of a corrugation crease in the limit of $w \to 0$. We define $\mathcal{H}_F^\rho(\rho(k, p))$ and $\mathcal{H}_F^\beta(\beta(k, p))$ as the stored energy functions associated to folding of the corrugation creases and the diagonal creases, respectively. As for the bending energy defined in Eq. (19), $\mathcal{H}_B^\theta(\theta(k, p))$ is the associated stored energy function of bending creases. We make the following basic assumptions:

$$\mathcal{H}_F^\beta \geq 0, \ \frac{\partial^2 \mathcal{H}_F^\beta}{\partial \beta^2} > 0, \ \frac{\partial \mathcal{H}_F^\beta}{\partial \beta}\bigg|_{\beta = \beta_0} = 0, \quad (20)$$

$$\mathcal{H}_F^\rho \geq 0, \ \frac{\partial^2 \mathcal{H}_F^\rho}{\partial \rho^2} > 0, \ \frac{\partial \mathcal{H}_F^\rho}{\partial \rho}\bigg|_{\rho = \rho_0} = 0, \quad (21)$$

$$\mathcal{H}_B^\theta \geq 0, \ \frac{\partial^2 \mathcal{H}_B^\theta}{\partial \theta^2} > 0, \ \frac{\partial \mathcal{H}_B^\theta}{\partial \theta}\bigg|_{\theta = 0} = 0. \quad (22)$$

We define the neutral angles of $\beta_0$ and $\rho_0$ as the folded angles at state $k = k_0$. The energy functions (18) and (19) allow us to interpret the mechanical behavior of the hypar pattern. Since the shape coefficient $k$ appears in Eqs. (18) and (19) in its quadratic form (i.e., $k^2$), we conclude that symmetry of system energy (with respect to $k$) exists for $k < 0$ and $k > 0$. Furthermore, for $0 < k < k_0$,

we find that $\partial E_T/\partial k < 0$ when $k \to 0$, and $\partial E_T/\partial k > 0$ when $k = k_0$. Therefore, there must exist a local minimum of $E_T$ at $0 < k^* < k_0$. Owing to the symmetry of system energy in $k$, there is another local minimum of $E_T$ at $-k_0 < -k^* < 0$. As a result, the system is guaranteed to exhibit bistability, and the two bistable states are symmetric. The reader is referred to Supplementary Note 3 for details of the derivation.

**Physical shape of folded hypar origami.** In reality, it is nearly impossible to confine isometric deformations while folding a hypar pattern due to the in-plane compliance of real materials. However, as thin elastic sheets usually deform in near-isometric states[37–39], we find that the analytical result, based on isometric folding, provides good approximations for the global and local geometries of a hypar origami made with real materials.

To investigate the subtleties of this near-isometric behavior, we fabricate two physical models with different thickness and capture their shapes using a 3D scanner. To visualize their in-plane and out-of-plane deformations, we compute the Gaussian curvature ($K$) and mean curvature ($H$) of the scanned surfaces[40]. For an initially flat sheet, in the small strain limit, the energy associated with stretching and bending increase with the magnitudes of $K$ and $H$, respectively[37,39]. The ratio of bending to stretching energy for a thin elastic sheet is proportional to $h^2$[39]. Therefore, thinner sheets favor less in-plane stretch than thicker sheets, as shown in Fig. 5a–d, where the thicker panels display larger $K$, while the thinner panels show larger $H$. In addition, stronger singular ridge effect[38,39] is found in the thinner origami. The twist of thinner panels creates local wrinkles near the ends of long panels, associated with zig-zag lines of Gaussian curvature concentration, as shown in Fig. 5c, d.

Closer examinations in Fig. 5e–i reveal that each panel displays a dominant diagonal about which the panel bends more than the other diagonal. Figure 5f, g is a schematic that shows the difference between an isometrically deformed origami panel and a non-isometrically deformed origami panel, assuming that straight creases remain straight. An isometrically deformed origami panel will display a single curvature because the Gaussian curvature is zero everywhere inside the panel. On the other hand, a non-isometrically deformed origami panel displays double curvature (negative Gaussian curvature) because this deformation mode stores less elastic energy. For the thinner panels as shown in Fig. 5h, by comparing the depth of bending (the black lines), we identify obvious dominant diagonals (AE and E′C) that align with the alternating asymmetric triangulation. In Fig. 5i, we also see that the dominant diagonals are AE and E′C for the thicker hypar model; however, the more in-plane deformation makes the dominant out-of-plane bending diagonals less distinguishable as in the thinner panels.

We collect positional information from the 3D images to compare with the analytical predictions in Fig. 2e. We sample coordinates of the mountain vertices to get estimates for the coefficient $k$. We then pick the middle points of square creases to form a zig-zag path (i.e., the green lines in Fig. 5e) to estimate the folding angles ($\rho$) of the corrugations.

**Folding and snapping by numerical simulations.** To study the mechanical behavior of the hypar origami considering non-isometric deformations, we conduct nonlinear structural analyses using the reduced order bar-and-hinge model (see Supplementary Fig. 6)[41,42]. Two numerical models are built for the cases of $h = 76.2 \,\mu m$ and $h = 127 \,\mu m$.

In the numerical simulation, we first fold up a flat pattern into a folded state, and then shift the neutral angles of the folding hinges accordingly, allowing the pattern to find a new equilibrium after

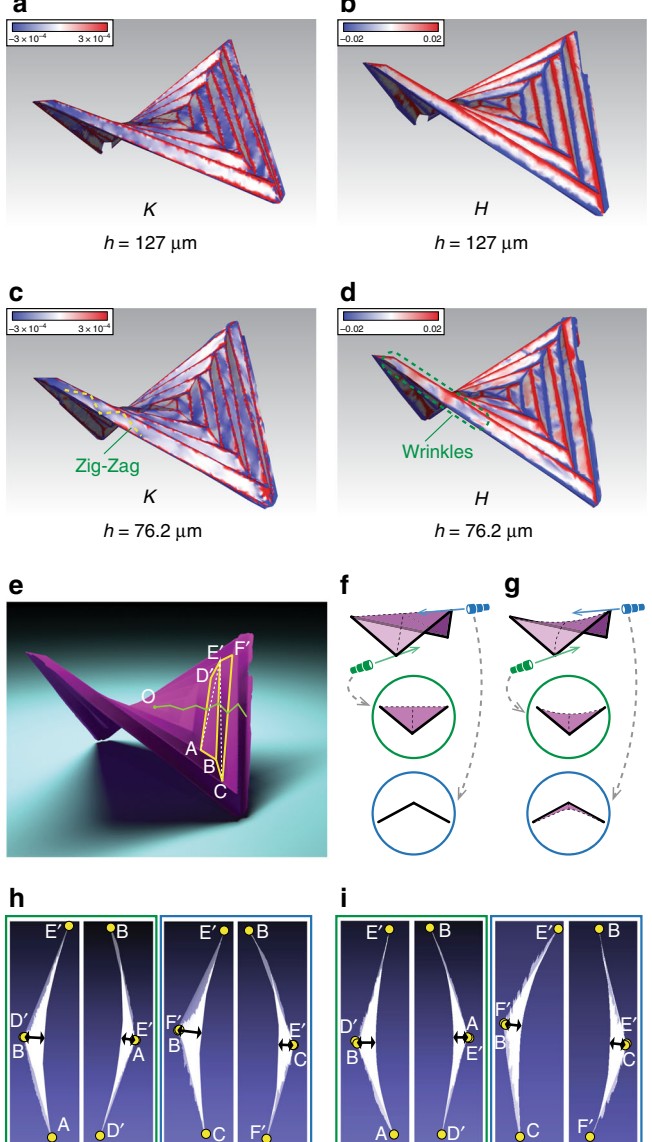

**Fig. 5** Three-dimensional images of two hypar origami made with Mylar sheets of different sheet thickness, $h = 127 \,\mu m$ (thick) and $h = 76.2 \,\mu m$ (thin). **a**, **c** Gaussian curvature ($K$) maps. **b**, **d** Mean curvature ($H$) maps. For both models, panel width $d = 8 \,mm$. Large $K$ and $H$ near the folded creases are truncated in the maps, allowing us to visualize the small curvature regions inside the panels. **e** The scanned 3D image of the thicker hypar origami model ($h = 127 \,\mu m$). The green line connects middles points of the long edges of the panels, which is used to estimate the folding angles ($\rho$) of the corrugations. **f** Schematics for three different views of an isometrically deformed panel: isometric projection and two projections looking through each diagonal respectively. The deformed shape displays a singly curved shape with curvature concentrated along one diagonal. **g** Schematics for different views of a non-isometrically deformed panel that involves in-plane stretching. The deformed shape displays a doubly curved shape with bending along both diagonals. There could be one dominant diagonal about which the panel bends more than the other. **h** Projections of panels ABE′D′ and BCF′E′ looking through the diagonals, from the thicker hypar origami model. **i** Projections of panels ABE′D′ and BCF′E′ looking through the diagonals, from the thinner hypar origami model

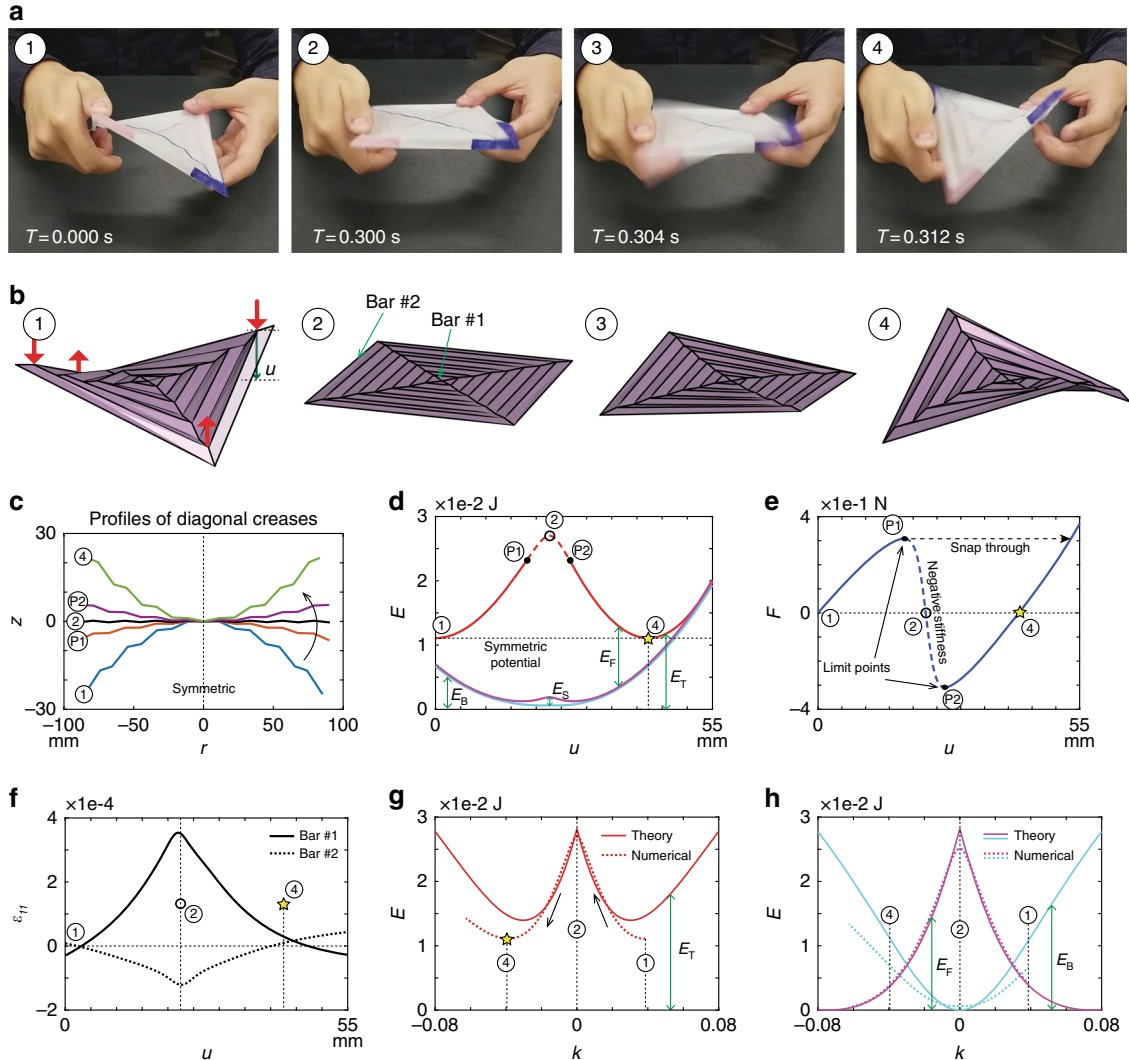

**Fig. 6 Snapping of the hypar origami. a** Snapshots picturing the snapping process of a Mylar-made physical model of thickness $h = 76.2\,\mu m$ (see Supplementary Movie 1). We show four frames along the process, in which $t$ indicates the frame time relative to the first one. **b** Frames taken from numerical simulation of the snapping process, using the thinner model ($h = 76.2\,\mu m$) as an example (see Supplementary Movie 2). To simulate the snapping, we apply equal magnitude forces following the red arrows. **c** The changing profile of one set of diagonal creases that approximates the curve $\xi$ with varying curvature and projection distance $r$. **d** Stored energy vs. displacement ($u$) plot. Contributions from three deformation modes: folding ($E_F$), bending ($E_B$), and stretching ($E_S$) are distinguished. **e** Force magnitude ($F$) vs. displacement ($u$) plot from numerical analysis, where $u$ is illustrated in **b**. **f** The Green-Lagrange strain ($\varepsilon_{11}$) in two bar elements. Bar #1 represents a central crease, and bar #2 represents an outermost panel edge, as indicated in **b**. Negative values refer to compression. **g** Comparison of the system total energy ($E_T$) as a function of $k$ computed by the analytical model and the numerical model. The black arrows indicate the loading direction. Notice that here the solid line refers to the analytical prediction. **h** Comparison of each individual energy component (i.e. $E_F$, $E_B$) computed by the analytical model and the numerical model. Notice that the analytical model ignores in-plane deformation and its associated energy $E_S$

releasing the applied folding forces (see Supplementary Fig. 7 and Supplementary Movie 3). Comparing the equilibrium configurations of both numerical models with the analytical prediction, we find good agreement as shown in Fig. 2e. Eigenvalue analysis on the stiffness matrix at the equilibrium configuration of the thinner model shows that the hypar folding mode possesses a much lower energy cost than other deformation modes, ~5% to the next smallest eigenvalue, which is an evident implication that the non-rigid hypar pattern has nearly a single degree of freedom. Zooming into each panel, both models display dominant bending diagonals forming the alternating asymmetric triangulation (see Supplementary Fig. 8), agreeing with the experiments.

The snapping between the two stable states is a rapid process, which happens with the blink of an eye (0.1–0.4 s[43]), as captured

in Fig. 6a. Our numerical simulation captures this bistable snapping in a quasi-static manner, and the results are presented in Fig. 6b–f. The change of stored energy during the snapping deformation is compared to the analytical prediction, i.e. Eqs. (18) and (19), in Fig. 6g, h, using the same constitutive models for folding and bending hinges as used in the bar-and-hinge model (see Supplementary Note 4). We obtain very nice agreement: the folding energy matches almost perfectly; the bending energy is overestimated, but it is because, by construction, the numerical model has richer kinematics than the analytical model, as the later ignores in-plane deformation and its associated energy $E_S$. The snapping process does not require the pattern to be completely flattened, owing to the presence of in-plane deformation. We observe severe tension in the central region and strong

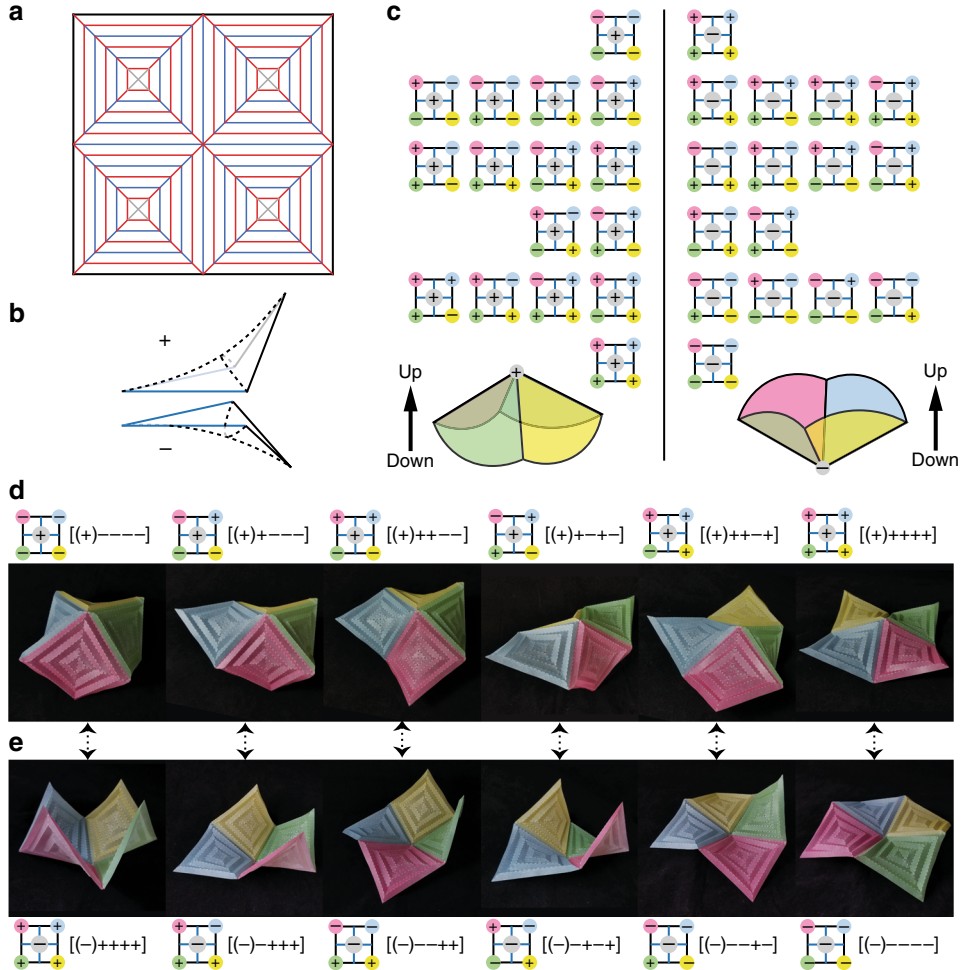

**Fig. 7** Multistable hypar origami tessellation built from a 2 × 2 array of hypar origami unit. **a** The crease pattern on a single sheet. **b** The convention of (+) and (−) states for each hypar origami unit. The blue edges are shared edges between adjacent units. **c** Complete chart of the 32 stable states. On the left side, there are 16 states when the middle vertex is in a pop-up state. On the right, the middle vertex is in a pop-down state. On either side, each row lists configurations that are identical to each other after rotations. **d** The six unique stable states, up to rotational symmetry, when the middle vertex is in a pop-up state. **e** The six unique stable states, up to rotational symmetry, when the middle vertex is in a pop-down state. The corresponding pairs of configurations in **d**, **e**, as indicated by the double arrows, lead to the same global geometry if one is flipped over (upside down). We can encode each state by five symbols, as labeled in **d**, **e**. The sign in the parenthesis indicates the state of the middle vertex, and the other four refer to the four hypar origami units. Since the hypar metasurface is rotationally symmetric, only the relative order of the last four signs affects the geometry. Accordingly, when all signs in a code become opposite, the global configuration is flipped over

compression at outer rims (see Fig. 6f). The outermost panels of a physical model often exhibit compressive buckling during the snapping, especially with relatively soft materials such as paper[34].

The bistable snapping of the hypar origami produces two stable states that are symmetric to each other, as the analytical model predicts. The mountain-valley assignment also remains the same before and after snapping. We observe that the angles of the folding hinges of the hypar origami remain the same at both stable states, as shown in Supplementary Fig. 8. However, each individual panel reverses its twisting directions during the snapping. On the contrary, most bistable origami structures display distinct global configurations at their bistable states, such as the square twist origami[22] and the Kresling pattern[19].

**Hypar tessellation with many stable states.** If we assemble several copies of the hypar pattern in a planar array, then after folding, we obtain origami metasurfaces that exhibit multiple stable states. In Fig. 7, we demonstrate this idea by assembling a 2 × 2 array of hypar patterns, known as the 4-hat[44], whose crease pattern is shown in Fig. 7a. This metasurface has 32 stable states,

which is doubled the binary combinations of bistable units (i.e., $2^4 = 16$). The reason is because when we put four hypar origami together, globally they form a vertex of positive Gaussian curvature in the center, as indicated by the gray circles in Fig. 7c, which can either pop-up (+) or pop-down (−). Hence, we create an origami metasurface that has $2(2^4) = 32$ stable states. Among the 32 stable states, 12 of them are unique up to rotational symmetry, and 6 of them are distinct in terms of approximate global shape. This idea can lead to programmable metasurfaces and metamaterials, which may have important applications in energy trapping[45], and micro electronic devices[7].

## Discussion

Our study shows that folding concentrically pleated squares produces shapes that asymptotically approach smooth hyperbolic paraboloids. Such a global saddle shape is strongly constrained by geometry and quite robust to some variations of the crease pattern (see Figs. 1 and 4). Implied by our study, a unique feature of the hypar origami is that throughout its folding process, the folded geometry is always a hyperbolic paraboloid, except for

different shallowness, which can be very useful for optical applications[46]. We further prove that the bistability of the hypar origami exist unconditionally to produce two symmetric stable configurations, when a few basic assumptions are satisfied. Our theoretical analysis is verified by experiments and numerical simulations.

Along with previous work on circular curved folds[12,13], our study proposes an analytical framework for homogenizing non-periodic local folds to establish a differential map of the global geometry, which can be used as a basis to investigate other corrugated origami shells, such as concentrically pleated patterns with polygonal outlines. Furthermore, we provide an example in the use of the hypar pattern as a bistable constituent in a multistable metasurface. In fact, recent papers[47,48] have pointed out an observable paradigm shift, away from avoiding instabilities to harnessing instabilities, which could be explored with the hypar origami. In summary, we offer an example of bistable behavior that emerges inherently from the geometry of folding.

## Methods

**Fabrication and testing of physical samples**. The two physical models are made of Mylar sheets with two different thicknesses ($h$): 127 μm (0.005 inch) and 76.2 μm (0.003 inch). We use Mylar sheets instead of regular paper because it has a more homogeneous and isotropic elastic behavior. The average modulus of elasticity of the Mylar material for all directions is $Y = 5$ GPa (725 ksi), and we assume the Poisson's ratio to be $v = 0.35$. The size of the models are characterized by panel width $d = 8$ mm. The predefined creases are perforated by slots, whose lengths add up to approximately one half of a crease. We use a Silhouette CAMEO machine (Silhouette America Inc., Utah) to prepare the perforated patterns. The two patterns are then gently hand folded to similar shapes. We use a hand-held 3D scanner (Artec Spider Scanner, Artec 3D, Luxembourg) that provides three-dimensional images with resolution up to 0.1 mm. In reference to Fig. 5, we crop one corrugation (two adjacent panels) from each of the scanned surfaces of the thicker and thinner model, and compute the average Gaussian curvatures inside the panels. The average Gaussian curvature of the thicker corrugation is equal to $-1.0135 \times 10^{-4}$, while the average Gaussian curvature of the thinner corrugation is $-0.9589 \times 10^{-4}$, i.e., a negative value with smaller magnitude, verifying that the thinner panels favor less in-plane stretch.

**Numerical simulations**. The numerical simulations are performed using the MERLIN software[42], which implements the bar-and-hinge model for nonlinear analysis of origami structures[41,42]. A nonlinear elastic formulation describes the constitutive behavior of each element in the model[42,49]. Please see the Supplementary Note 4 for more details on the bar-and-hinge model and the nonlinear elastic formulation. The nonlinear equilibrium problem is solved by the Modified Generalized Displacement Control Method, which is able to trace highly nonlinear equilibrium paths[50]. We discretize a quadrilateral panel into four triangles, and represent the origami behavior by capturing three essential deformation modes: folding, panel bending, and stretching.

## Data availability

The authors declare that the data generated or analyzed during this study are included in this article and its supplementary files.

## Code availability

The MERLIN software (MATLAB code) used for the numerical simulations in this article is available at http://paulino.ce.gatech.edu/software.html.

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

## Acknowledgements
We thank the support from the US National Science Foundation (NSF) through grant no.1538830, the China Scholarship Council (CSC), and the Raymond Allen Jones Chair at Georgia Tech. The authors would like to extend their appreciation to Mrs. Emily D. Sanders for helpful discussions which contributed to improve the present work.

## Author contributions
K.L., T.T. and G.H.P. designed the research. K.L. performed theoretical development, experiments, and numerical simulations. T.T. and G.H.P. provided guidance throughout the research. All the authors participated in manuscript writing and reviewed the manuscript.

## Additional information

**Competing interests:** The authors declare no competing interests.

