## [Peer Review File · Nature Communications]

Reviewers' comments:

Reviewer #1 (Remarks to the Author):

This paper principally investigates the shape of the pleated hyperbolic paraboloid (hypar) origami. As the name suggests the shape appears to approximate a hyperbolic paraboloid. The study derives a connection between the discrete pleats which constitute the origami and a smooth ruled surface. In the limit of a large number of pleats the authors show the pleats approximate the rulings of a ruled surface. This ruled surface is found to be a hyperbolic paraboloid. The authors also show that this shape is relatively insensitive to inaccuracies in the folding pattern.

Numerical simulations of the folding of the origami hypar are performed using a bar-and-hinge model and compared to the analytical results showing good agreement. Additionally, experiments are performed by folding models made from Mylar. The shape of these models is extracted using 3D scanning and compared to the numerical and analytical results showing good agreement.

A numerical analysis, using the bar and hinge model, of the bistable behaviour of the origami hypar is also included. Finally a demonstration of the multistability of a tessellation of hypar origami units is provided.

This work well written, interesting and makes a valuable contribution to the analysis of the shape of origami models.

There are some minor issues which the authors should address before publication:

- Pg. 2 '... have the same shape but opposite orientations (see Fig. S1(e))'
 - o I don't see this shown in Fig. S1
- I found the opening angles ψ and ψ' hard to understand from Fig. 2. Could this be shown more clearly or explained in more detail? Perhaps they could be added to Fig. S1?
- I found Fig. 5(f-i) difficult to visualise what orientation is being shown. Could the vertex labels (A,B,C,D',E',F') from Fig. 5(e) be used in Fig. 5(f-g) to help the reader orient the diagrams to the overall shape. Are the views in Fig. 5(f-g) the same as shown in Fig. 5 (h-i)?
- Pg.8 1st full paragraph: 'As we can observe from Fig. S4...' It doesn't appear that Fig. S4 exists.
- The bistability part of this paper does not seem to be strongly connected with the main results regarding the analytical shape. The bistability of the origami hypar is known and has been studied recently by Filipov & Redouley (Extreme Mechanics Letters, 25, 2018) for example. Can your analytical results show that both stable states are hyperbolic paraboloids despite the fold angles remaining the same?
- In Supplementary material pg 6. 'Now we multiply both sides by $(eac/|eac|)$ '. Both sides of what equation? S20?

Some minor typographical issues that the authors may wish to address:

- Pg. 4. ' as a result of Eq. (3) being a necessary, but not sufficient, condition' in 1st paragraph of The Unexpected Solution repeated in second paragraph 'Eq. (3) is only a necessary condition to the compatibility constraints.'
- Caption of Fig. 3 'Detailed constriction' should be 'Detailed construction'?
- Pg. 4 1st Paragraph of Invariance of the analytical limit: 'a serendipity finding' should be 'a serendipitous finding'
- Supplementary materials pg 2. 1st sentence: 'Fig.S1(e) ... Fig. S1(f)' should be: Fig.S1(d)...Fig.S1(e)?
- Supplementary materials pg 2. 2nd sentence 'Although we find that both triangulation schemes...' repeated in last paragraph of pg 2.
- Movie captions are labelled S1...S3 which conflicts with supplementary materials Figs S1...S3.

Reviewer #2 (Remarks to the Author):

This manuscript is concerned origami model of the hyperbolic paraboloid, called hyper, well know for its aesthetically pleasing geometry as well as interesting mechanical behavior. The authors present an analytical description of kinematics of the this structure, which is comprised of folding concentric squares with alternating mountains valleys. Through a homogenization procedure, the

authors show that the kinematic equations yield the global hyperbolic paraboloid geometry. Following the study on the geometry, the authors suggest a bar-and-hinge model in order to understand the mechanics. An experimental observation of hyper's bistable nature is also explored by the authors. This last study leads to the claim that this origami pattern can be used to build mechanical metamaterials.

The topic of the manuscript is relevant for a diverse variety of members in the scientific community—namely engineers, physicists, and mathematicians. The authors' contribution appears plausible. There are, however, many aspects of their presentation that could be improved. The two main aspects of their work, kinematical description of the global shape and mechanics of bistability, were not very well connected. The numerical study of the bistability was quite terse and the lack of parametric exploration from the numerical model left many questions unexplored. For instance, how could we understand the influence of functionally graded hyper pattern in the question of stability? Very little discussion on the type of crease energy was provided, hence its effect on stability was also unexplored.

The following are some other suggestions and questions to be addressed:

- 1) The abstract is a bit unorganized. It needs more focus in order to convey a clear message.
- 2) In the second paragraph, the sentence "Within this category [27], the hyper differentiates itself by displaying bistability with symmetric configurations." This is not only a pattern that exhibits this characteristic. Much simpler patterns present bistability and they are not necessarily non-rigid. For example see Adda-Bedia et al. Also, there seems to be the case that the non-periodicity presents no particular advantage. One could break down this behavior down to the local level of the one vertex and a set of creases, as it has already been demonstrated by Adda-Bedia and Seffen. Here are the references:
[1] F. Lechenault and M. Adda-Bedia, Generic Bistability in Creased Conical Surfaces, Phys. Rev. Lett. 115, 235501 (2015)
[2] Walker, Martin G., and Keith A. Seffen. "On the shape of bistable creased strips." Thin-Walled Structures 124 (2018): 538-545.
- 3) In the beginning of the second column, first page, the authors layout three questions, of which only the last one remains unexplored. Since the other questions have already been investigated, it would be instructive if the authors made their points more precise in comparison with Demaine et al.
- 4) In the sentences "homogenizing local folds to establish a differential map of the global geometry." Maybe worth mentioning such a task has been carried out for concentric circles, which was another open question left by Demaine et al. and solved by Dias and Santangelo: Dias, Marcelo A., and Christian D. Santangelo. "The shape and mechanics of curved-fold origami structures." EPL (Europhysics Letters) 100.5 (2012): 54005.
- 5) Right after equation (1), the authors mention "isometric deformations". Here it is not clear what the authors mean by isometric deformations, since a twisted state could still be isometric to a plane. Are the authors referring to a more constraining situation, i.e. rigidity?
- 6) When writing $(d/L)=1/2n$, perhaps the authors could use an extra parenthesis in order to make clear where the variable n sits, i.e. $(d/L)=1/(2n)$.
- 7) In the computation of Gaussian and Mean curvature from the scanner data, it is difficult to judge the results due to the relatively low quality of the calculated curvatures. Is there any way the authors could show more quantitative results?
- 8) In page 6, section "Folding process by numerical simulations", in the sentence "A nonlinear elastic formulation describes the constitutive behavior of each element in the model...", what do

the authors mean by "nonlinear elastic formulation" ? Here, they are giving the impression the choice of constitutive law is nonlinear. Is this the case? If so, what nonlinear model is here used? If not, please clarify.

Response to Reviewers

We are grateful to the reviewers for their feedback and helpful critique. The editor's and reviewers' comments have been addressed in the manuscript, and the revisions are summarized below. Our response to the reviewers are written in purple below. Portions of the text that have been modified are colored in red in the revised manuscript.

Reviewer #1:

This paper principally investigates the shape of the pleated hyperbolic paraboloid (hypar) origami. As the name suggests the shape appears to approximate a hyperbolic paraboloid. The study derives a connection between the discrete pleats which constitute the origami and a smooth ruled surface. In the limit of a large number of pleats the authors show the pleats approximate the rulings of a ruled surface. This ruled surface is found to be a hyperbolic paraboloid. The authors also show that this shape is relatively insensitive to inaccuracies in the folding pattern.

Numerical simulations of the folding of the origami hypar are performed using a bar-and-hinge model and compared to the analytical results showing good agreement. Additionally, experiments are performed by folding models made from Mylar. The shape of these models is extracted using 3D scanning and compared to the numerical and analytical results showing good agreement.

A numerical analysis, using the bar and hinge model, of the bistable behaviour of the origami hypar is also included. Finally, a demonstration of the multistability of a tessellation of hypar origami units is provided. This work well written, interesting and makes a valuable contribution to the analysis of the shape of origami models.

→ **Response:** We thank the reviewer for the compliment about our work.

There are some minor issues which the authors should address before publication:

1. Pg. 2 ‘... have the same shape but opposite orientations (see Fig. S1(e))’ I don’t see this shown in Fig. S1

→ **Response:** We thank the reviewer for pointing this mistake out. We have corrected the reference to the figure, which should be “Fig. 1(d).”

2. I found the opening angles ψ and ψ' hard to understand from Fig. 2. Could this be shown more clearly or explained in more detail? Perhaps they could be added to Fig. S1?

→ **Response:** We have made a separate figure, Fig. S2, to illustrate the geometric definitions of angles ψ and ψ' .

The ψ angle is the angle between e_{AC} and e_{DF} . After twisting of the corrugation, D and F displace to new positions, and they are labeled as D' and F', respectively. The angle between e_{AC} and $e_{D'F'}$ is denoted by ψ' . The relationship between ψ , ψ' , and γ is illustrated in part (c) of Fig. S2. We appreciate the question from the reviewer and we believe that the information added in conjunction to the new Figure (Fig. S2) clarifies the notation used throughout the paper.

3. I found Fig. 5(f-i) difficult to visualise what orientation is being shown. Could the vertex labels (A,B,C,D',E',F') from Fig. 5(e) be used in Fig. 5(f-g) to help the reader orient the diagrams to the overall shape. Are the views in Fig. 5(f-g) the same as shown in Fig. 5 (h-i)?

→ **Response:** Fig. 5 (f-g) are not the same as shown in Fig. 5 (h-i). In fact, Fig. 5 (f-g) is a schematic that shows the difference between an isometrically deformed origami panel and a non-isometrically deformed origami panel, assuming that straight creases remain straight. An isometrically deformed origami panel will display a single curvature, because the Gaussian curvature is zero everywhere inside the panel. On the other hand, a non-isometrically deformed origami panel displays double curvature (negative Gaussian curvature), because this deformation mode stores less elastic energy. We use Fig. 5 (f-g) to demonstrate how a deformed origami panel look like under isometric and non-isometric deformations, if we inspect through its diagonals. We hope that our explanation is helpful.

To accommodate the reviewer's comment, we have made the following changes to the main text (Page 6): Fig. 5 (f-g) is a schematic that shows the difference between an isometrically deformed origami panel and a non-isometrically deformed origami panel, assuming that straight creases remain straight. An isometrically deformed origami panel will display a single curvature because the Gaussian curvature is zero everywhere inside the panel. On the other hand, a non-isometrically deformed origami panel displays double curvature (negative Gaussian curvature) because this deformation mode stores less elastic energy.

4. Pg.8 1st full paragraph: 'As we can observe from Fig. S4...' It doesn't appear that Fig. S4 exists.

→ **Response:** We are sorry for the unintended typo. This figure is now labeled as Fig. S7 in the new version of Supplementary Information (SI) Text part.

5. The bistability part of this paper does not seem to be strongly connected with the main results regarding the analytical shape. The bistability of the origami hyper is known and has been studied recently by Filipov & Redoutey (Extreme Mechanics Letters, 25, 2018) for example. Can your analytical results show that both stable states are hyperbolic paraboloids despite the fold angles remaining the same?

Response: We agree with the critique of the reviewer. Therefore, we have created a theoretical model to describe the energy of the folded hyper system utilizing homogenized geometry. The theory is based on the assumption that there exist convex stored energy functions for folding and bending hinges with different neutral states, i.e. the neutral angles of folding hinges correspond to a folded state, while the neutral angles of bending hinges correspond to the flat (unfolded) state. We have proven that the bistability of the hyper pattern exists unconditionally, i.e. regardless of the specific constitutive models, and crease offsets. In addition, we have shown that the two stable hyperbolic paraboloid states are symmetric wrt 90 degree rotations. Finally, we have compared the analytical (homogenized) prediction of the system energy with the numerical model and obtained very nice agreement: *the folding energy matches almost perfectly; the bending energy is overestimated, but this happens because, by construction, the numerical model has richer kinematics than the theoretical model.* We believe that the reviewer will appreciate this new development.

Please see SI Section III for detailed derivation of the new theory.

Moreover, we have added a new Section to the manuscript, which is titled “**Connecting geometry with mechanics.**”

6. In Supplementary material pg 6. ‘Now we multiply both sides by $(e_{AC}/|e_{AC}|)$ ’. Both sides of what equation? S20?

→ **Response:** The reviewer is correct. Here we mean both sides of Eq. (S20). We have made the following changes in the SI:

Now we multiply both sides of Eq. (S20) by $e_{AC}/|e_{AC}|$.

Some minor typographical issues that the authors may wish to address:

7. Pg. 4. ‘as a result of Eq. (3) being a necessary, but not sufficient, condition’ in 1st paragraph of The Unexpected Solution repeated in second paragraph ‘Eq. (3) is only a necessary condition to the compatibility constraints.’

→ **Response:** We thank the reviewer for pointing this out. We have deleted the first appearance of this sentence.

8. Caption of Fig. 3 ‘Detailed constriction’ should be ‘Detailed construction’?

→ **Response:** This is a typo. To be clearer, we have changed the phrase to “Detailed structure.”

9. Pg. 4 1st Paragraph of Invariance of the analytical limit: ‘a serendipity finding’ should be ‘a serendipitous finding’

→ **Response:** Thanks. We have made this change as the reviewer suggested.

10. Supplementary materials pg 2. 1st sentence: ‘Fig. S1(e) ... Fig. S1(f)’ should be: Fig.S1(d)...Fig.S1(e)?

→ **Response:** Thanks. The references to figures are not correct here. We have corrected the errors.

11. Supplementary materials pg 2. 2nd sentence ‘Although we find that both triangulation schemes...’ repeated in last paragraph of pg 2.

→ **Response:** We thank the reviewer for pointing this out. We have deleted the second appearance of this sentence.

12. Movie captions are labelled S1...S3 which conflicts with supplementary materials Figs S1...S3.

→ **Response:** We have relabeled the movies as Movie 1, 2, and 3, removing the “S” letter.

Reviewer #2:

This manuscript is concerned origami model of the hyperbolic paraboloid, called hyper, well know for its aesthetically pleasing geometry as well as interesting mechanical behavior. The authors present an analytical description of kinematics of the this structure, which is comprised of folding concentric squares with alternating mountains valleys. Through a homogenization procedure, the authors show that the kinematic equations yield the global hyperbolic paraboloid geometry. Following the study on the geometry, the authors suggest a bar-and-hinge model in order to understand the mechanics. An experimental observation of hyper's bistable nature is also explored by the authors. This last study leads to the claim that this origami pattern can be used to build mechanical metamaterials.

The topic of the manuscript is relevant for a diverse variety of members in the scientific community—namely engineers, physicists, and mathematicians. The authors contribution appears plausible. There are, however, many aspects of their presentation that could be improved. The two main aspects of their work, kinematical description of the global shape and mechanics of bistability, were not very well connected. The numerical study of the bistability was quite terse and the lack of parametric exploration from the numerical model left many questions unexplored. For instance, how could we understand the influence of functionally graded hyper pattern in the question of stability? Very little discussion on the type of crease energy was provided, hence its effect on stability was also unexplored.

→ **Response:** We thank the reviewer for the positive comments about our work. We agree with the reviewer that the connection between the two main aspects of our work (i.e. kinematic description of global shape and mechanics of bistability) is, unfortunately, weak. Hence, to address the reviewer's question, we have now created a theoretical model to describe the energy of the folded hyper system based on the homogenized geometry. We have compared the analytical (homogenized) prediction of the system energy with the numerical model and obtained very nice agreement: *the folding energy matches almost perfectly, the bending energy is overestimated but it is because, by construction, the numerical model has richer kinematics*. We have also proven that the bistability of the hyper pattern exists unconditionally, regardless of the specific constitutive model of bending hinges, folding hinges, and creases offsets. That is, the functionally graded hyper patterns are also guaranteed to be bistable. We have shown that the two stable states are symmetric hyperbolic paraboloid. We believe that the reviewer will appreciate this new development.

Please see SI Section III for detailed derivation of the new theory.

Moreover, we have added a new Section to the manuscript, which is titled “**Connecting geometry with mechanics.**”

The following are some other suggestions and questions to be addressed:

1. The abstract is a bit unorganized. It needs more focus in order to convey a clear message.

→ **Response:** We have substantially rewritten the abstract to address the reviewer's comment. The new abstract reads:

Origami offers an avenue to program three-dimensional shapes via scale-independent and non-destructive fabrication. While such programming has focused on the geometry of a tessellation in a single transient state, here we provide a complete description of folding smooth saddle shapes from concentrically pleated squares. When the offset between square creases of the pattern is uniform, it is known as the pleated hyperbolic

paraboloid (hypar) origami. Despite its popularity, much remains unknown about the mechanism that produces such aesthetic shapes. We show that the mathematical limit of the elegant shape folded from concentrically pleated squares, with either uniform or non-uniform (e.g. functionally graded, random) offsets, is invariantly a hyperbolic paraboloid. Using our theoretical model, which connects geometry to mechanics, we prove that a folded hypar origami exhibits bistability between two symmetric configurations. Further, we tessellate the hypar origami and harness its bistability to encode multi-stable metamaterials with programmable non-Euclidean geometries.

2. In the second paragraph, the sentence “Within this category [27], the hypar differentiates itself by displaying bistability with symmetric configurations.” This is not only pattern that exhibits this characteristics. Much simpler patterns present bistability and they are not necessarily non-rigid. For example see Adda-Bedia et al. Also, there seems to be the case that the non periodicity presents no particular advantage. One could break down this behavior down to the local level of the one vertex and a set of creases, as it has already been demonstrated by Adda-Bedia and Seffen. Here are the references:
[1] F. Lechenault and M. Adda-Bedia, Generic Bistability in Creased Conical Surfaces, Phys. Rev. Lett. 115, 235501 (2015)
[2] Walker, Martin G., and Keith A. Seffen. "On the shape of bistable creased strips." Thin-Walled Structures 124 (2018): 538-545.

→ **Response:** Here the main difference between the hypar origami and other bistable/multistable origami is that the two stable configurations of the hypar origami are symmetric to each other, in addition to the symmetric crease pattern. Although the two examples provided by the reviewer possess symmetric crease patterns, they do not produce symmetric stable states. However, we thank the reviewer for pointing out these two important pieces of work – we have cited them to provide proper credit to the pertinent literature (see references [23] and [25] and related comments in the first paragraph of the manuscript). Moreover, we have modified the text, as follows, in order to make our idea clearer.

Page 1: The hypar pattern also exhibits interesting mechanical properties, such as structural bistability [33,34]. An interesting and quite exclusive fact is that the two stable configurations of a hypar origami are symmetric to each other, which makes it a promising platform for multi-functional devices or metamaterials.

3. In the beginning of the second column, first page, the authors layout three questions, of which only the last one remains unexplored. Since the other questions have already been investigated, it would instructive if the authors made their points more precise in comparison with Demaine et al.

→ **Response:** Thanks for the comment. We have modified the questions such that they directly reflect our novel contributions. Thus, the posed questions now read as follows:

Page 1: In an attempt to gain a deeper theoretical understanding of the hypar origami and use it for engineering applications, we aim to answer the following questions: *Can we prove, theoretically, that the actual folded shape of the hypar origami is a hyperbolic paraboloid? How does the local deformation of each panel relates to the global shape of the pattern? What are the conditions for the bistability to exist? How can we use the hypar pattern to create metamaterials?*

Notice that now we pose this 4 relevant questions and we answer them. These questions provide the motivation to the work.

4. In the sentences “homogenizing local folds to establish a differential map of the global geometry.” Maybe worth mentioning such task has been carried out for concentric circles, which was another open question

left by Demaine et al. and solved by Dias and Santangelo: Dias, Marcelo A., and Christian D. Santangelo. "The shape and mechanics of curved-fold origami structures." EPL (Europhysics Letters) 100.5 (2012): 54005.

→ **Response:** We agree with the reviewer's suggestion. Thus we have added the aforementioned reference to provide proper credit to the pertinent literature. In addition, we added some comments on this regard to the main text:

Page 10: Along with previous work on circular curved folds [12, 13], our study proposes an analytical framework for homogenizing non-periodic local folds to establish a differential map of the global geometry, which can be used as a basis to investigate other corrugated origami shells, such as concentrically pleated patterns with polygonal outlines.

5. Right after equation (1), the authors mention "isometric deformations". Here it is not clear what the authors mean by isometric deformations, since a twisted state could still be isometric to a plane. Are the authors referring to a more constraining situation, i.e. rigidity?

→ **Response:** We agree that this statement is a bit confusing. We should have mentioned that we are considering the case when all the creases remain straight after deformation. Under such assumption, to achieve isometric deformation, the trapezoidal panels must bend along one of its diagonals.

Thus, we have added the following text in the manuscript in order to clarify our assumptions:

Page 2: To analytically describe the geometry of each deformed corrugation, we assume that the deformations are isometric, and straight creases remain straight, which requires the introduction of at least one additional diagonal pleat in each trapezoidal panel to triangulate the pattern [35].

6. When writing $(d/L)=1/2n$, perhaps the authors could use an extra parenthesis in order to make clear where the variable n sits, i.e. $(d/L)=1/(2n)$.

→ **Response:** We thank the reviewer for giving this suggestion. We added parenthesis to the term $(2n)$.

7. In the computation of Gaussian and Mean curvature from the scanner data, it is difficult to judge the results due to the relatively low quality of the calculated curvatures. Is there any way the authors could show more quantitative results?

→ **Response:** The main goal of the curvature analyses is to provide some insight into the deformed shapes of panels in actual (i.e. physical) hyper origami models. We wanted to show that the actual panels are not deformed isometrically, and the thinner the panel, the more isometric the deformation – qualitatively the Figures illustrate that point. However, to address the reviewer's comment, we have cropped one corrugation (two adjacent panels) from each of the scanned surfaces of the thicker and thinner model, and computed the average Gaussian curvatures. The average Gaussian curvature of the thicker corrugation equals to $-1.0135e-04$, while the average Gaussian curvature of the thinner corrugation equals to $-0.9589e-04$, indicating that the thinner panels favor less in-plane stretches.

We have added the following sentence to the main text (Page 10) to provide a quantitative measurement of the curvatures:

In reference to Fig.5, we crop one corrugation (two adjacent panels) from each of the scanned surfaces of the thicker and thinner model, and compute the average Gaussian curvatures. The average Gaussian curvature of

the thicker corrugation equals to $-1.0135e-04$, while the average Gaussian curvature of the thinner corrugation reads as $-0.9589e-04$, a negative value with smaller magnitude, verifying that the thinner panels favor less in-plane stretches.

As a final comment related to the reviewer's question, we would like to remark that, in terms of the accuracy of the computed curvatures, we are constrained by the resolution of our 3D scanner, which is 0.1mm. In addition, the physical models are folded by hand. Since the hyper origami is a non-rigid foldable pattern, the folding process is likely to introduce imperfections. Unfortunately, with our current access to scanning and manufacturing technology, it would be difficult for us to improve the accuracy of the curvature measurements further.

8. In page 6, section "Folding process by numerical simulations", in the sentence "A nonlinear elastic formulation describes the constitutive behavior of each element in the model...", what do the authors mean by "nonlinear elastic formulation"? Here, they are giving the impression the choice of constitutive law is nonlinear. Is this the case? If so, what nonlinear model is here used? If not, please clarify.

→ **Response:** The "nonlinear elastic formulation" here refers to the one established by our previous work (K. Liu and G. H. Paulino. Nonlinear mechanics of non-rigid origami: an efficient computational approach. Proceedings of the Royal Society A, 473:20170348, 2017), which is reference [42] in the manuscript. This elastic formulation considers both material and geometric nonlinearity. In this paper, the bending and folding hinges are assigned a linear constitutive model (associated to rotational degree of freedom), and the bar elements, which simulate in-plane stretching and shearing, are assigned a nonlinear hyperelastic constitutive model. Detailed description can be found in **Section IV of the Supplementary Information**.

REVIEWERS' COMMENTS:

Reviewer #1 (Remarks to the Author):

The authors have addressed all my concerns.

Reviewer #2 (Remarks to the Author):

The authors have addressed all my remarks and I'm satisfied with their manuscript. Hence, I recommend their work for publication.

Response to Reviewers' comments

REVIEWERS' COMMENTS:

Reviewer #1 (Remarks to the Author):

The authors have addressed all my concerns.

→ Response:

We thank the reviewer for recommending publication of our paper.

Reviewer #2 (Remarks to the Author):

The authors have addressed all my remarks and I'm satisfied with their manuscript. Hence, I recommend their work for publication.

→ Response:

We thank the reviewer for recommending publication of our paper.